# Induced Zinc Loss Produces Heterogenous Biological Responses in Melanoma Cells

**DOI:** 10.3390/ijms23158312

**Published:** 2022-07-27

**Authors:** Emil Rudolf, Kamil Rudolf

**Affiliations:** Department of Medical Biology and Genetics, Faculty of Medicine in Hradec Kralove, Charles University, Zborovska 2089, 500 03 Hradec Kralove, Czech Republic; rudolfk@seznam.cz

**Keywords:** free zinc, melanoma, chelation, cell death, premature senescence

## Abstract

Zinc levels in serum and/or tissue are reported to be altered in melanoma with unknown effects on melanoma development and biology. The purpose of this study was to examine the effects of acute chelation of free intracellular zinc pools in melanoma cell lines Bowes and A375, as well as selected melanoma tissue explants with high or low intracellular free zinc. Zinc chelating agent TPEN at the concentration of 25 µM was employed during 48 h, which significantly reduced intracellular free zinc while decreasing melanoma cell proliferation, inducing G1/S arrest and cell damage leading to mitochondrial, caspase-dependent apoptosis. Chelation of free zinc was also associated with increased generation of superoxide in cell lines but not marked lysosomal membrane damage. Conversely, melanoma explant cultures mostly displayed time-dependent loss of lysosomal membrane integrity in the presence of slowly growing superoxide levels. Loss of free zinc-dependent p53 activity was similarly disparate in individual melanoma models. Surviving melanoma cells were arrested in the cell cycle, and varying proportions of them exhibited features characteristic of premature senescence, which increased in time despite zinc reloading. The present results show that melanoma cells with varying free zinc levels respond to its acute loss in a number of individual ways, reflecting activated mechanisms including oxidative stress, lysosomal damage, and p53 activity leading to heterogenous outcomes including cell death, transient, and/or permanent cell cycle arrest and premature senescence.

## 1. Introduction

Several trace elements, including iron, copper, manganese, and zinc, are known to be essential for the growth and development as well as the homeostasis of most living organisms, including humans. Their importance for the structure and function of living matter has been consistently demonstrated at molecular, cellular, tissue, and organismal levels. In the human body, zinc is the second most abundant trace element with structural, regulatory, and catalytic functions underlying a wide range of biological processes such as integrity and expression of genetic information, signaling, redox homeostasis, cell proliferation, senescence, and demise [1]. Although its presence in an organism is universal, concrete zinc levels in individual intracellular and extracellular compartments significantly differ, ranging from pM or low nM to high µM concentrations [2]. In human cells, zinc is mostly bound by various substrates, including proteins and peptides, and its availability for interaction with other intracellular targets (compartments or individual molecules) is thus limited. Still, a very small fraction of intracellular zinc exists in free form; i.e., relatively weekly bound to low molecular weight ligands or not bound at all and these zinc ions act as second messengers or transcription regulators. To maintain the stability and proper compartmentalization of intracellular zinc levels, various zinc-specific transport, sequestering, buffering and mulling mechanisms are present [3]. The importance of zinc in the mentioned physiological processes and its regulation is further demonstrated by numerous reports showing a causal linkage between acute or chronic changes in zinc levels and various pathologies, including the development of neoplasia [4].

While generally acknowledged, the role of zinc in carcinogenesis remains somewhat controversial and continues to be not entirely elucidated. Epidemiological evidence indicates that both zinc deficiency as well as excess contribute to an increased risk of tumor development. This observation is significantly underscored and reflected by reported altered levels of serum zinc in various solid and hematological malignancies (breast, brain, liver, prostate, pancreas, skin, as well as leukemia [5]). Still, both increased or decreased serum zinc levels do not entirely correlate with zinc concentrations in particular malignant cells as reported in case of breast, prostate, or lung cancers, likely reflecting changes in intratumor/intracellular zinc management [6]. 

Skin melanoma is the least frequent but most dangerous type of skin cancer, with steadily increasing worldwide incidence. While well treatable at initial stages, advanced stages of this condition show a strong propensity for aggressive growth, chemoresistance, and systemic spread. Malignant melanoma cells are known to be genetically as well as phenotypically heterogeneous, which contributes to their functional plasticity, survival, and aggressiveness. Despite their heterogeneous nature, several key molecular signatures, including mutations in *BRAF*, *NRAS,* and *NF1* genes and dysregulation in related signaling pathways such as BRAF-MEK, PI3K/PTEN, or c-KIT cascades, have been identified [7]. Furthermore, other processes, as well as concrete cellular compartments, have been found to be involved in melanoma biology [8,9,10]. In addition, it has recently been found that trace element copper is involved in melanoma biology since it is required for BRAF-MEK signaling and when cellular access to it is limited, melanoma cell growth is markedly decreased [11,12]. Unlike copper, however, no such specific relationship has been reported for zinc in melanoma. Still, several indices exist, such as overexpression of zinc-binding metallothioneins or increased sensitivity of melanoma cells to elevated zinc concentrations [13], which suggest potentially important changes in zinc management of melanoma cells and their dependence on it. The present study was conducted to investigate the effects of induced zinc loss on survival and specific biological responses of melanoma cells. 

## 2. Results

### 2.1. Proliferation and Zinc Content in Human Melanoma Cells

Stabilized melanoma cell lines Bowes and A375, as well as two melanoma explant cultures (labeled M1 and M2—prepared from the specimens of patients with advanced malignancy state as described before), were initially studied to determine their proliferation rates and total as well as free zinc content. All examined melanoma cells showed a time-dependent growth and proliferation, which were fully comparable with the exception of the M2 explant culture, whose reproduction rate was considerably higher within 48 h interval (Figure 1A). Total intracellular zinc content ranged from approximately 0.62 (Bowes) to 0.96 (M2) µg/mg of protein in individual melanoma cells, being generally higher in explant melanoma cultures as compared to stabilized lines Bowes and A375 (Figure 1B). Free zinc pools differed significantly between individual melanoma cells too. Both Bowes and A375 cells had similar free zinc levels (equivalent to approximately 900 light units), whereas explant melanoma cultures showed markedly lower free zinc levels (M1—equivalent to approximately 430 light units) or significantly higher free zinc levels (M2—about 1800 light units) (Figure 1C). 

### 2.2. Effects of Zinc Sequestration on Proliferation and Free Zinc in Human Melanoma Cells

The effect of zinc-chelating agent TPEN on growth and proliferation as well as free zinc in the employed melanoma cell lines Bowes and A375, and human melanoma explant cultures M1 and M2 was tested. TPEN was used in a concentration range of 5–100 µM, reflecting previously published evidence. All tested TPEN concentrations, save for the lowest one (5 µM), reduced proliferation as well as free zinc pools in both Bowes and A375 cells during 48 h of treatment (Figure 2A–C). A similar effect was observed in the case of M1 and M2 explant melanoma cultures; however, the efficiency of TPEN in reducing proliferation and free zinc pools was greatly enhanced in M2 cells, where even the lowest TPEN concentration (5 µM) had a significant inhibiting effect towards proliferation and free zinc content (Figure 3A–C).

### 2.3. Depletion of Intracellular Zinc Reduces S-Phase Fraction of Human Melanoma Cells

Compared to controls, treatment of both human melanoma stabilized cells/cultures induced a significant reduction in the S-phase fraction cells, peaking at 48 h of TPEN exposure and being most obvious in M2 culture (Figure 4).

### 2.4. Chelation of Intracellular Zinc in Melanoma Cells Induces Cell Damage/Death

Previous concentration-dependent experiments demonstrated that 25 µM TPEN efficiently suppresses proliferation and significantly reduces intracellular free zinc levels in both stabilized melanoma cell lines and explant melanoma cultures. Thus treated cells/cultures were next analyzed morphologically and biochemically to determine whether the observed suppressive effects of TPEN were associated with cell damage/death. Time-lapse phase contrast microscopy revealed that already at 24 h of treatment, about 40% of cells of Bowes and A375 lines showed altered appearance, which in most cases included loss of adherence, rounding, and membrane blebbing. Further analyses confirmed that most of these cells had intact membranes corresponding to apoptotic modality, whereas only a minor fraction of them presented with necrotic or other phenotypes (Figure 5A–F). Similar results occurred in the case of explant melanoma cultures; however, the rate of morphologically/biochemically changed cells was significantly higher (up to 80%). In addition, unlike in cell lines, the apoptotic phenotype was present in about 50% of cells only, while necrotic and other phenotypes were roughly proportional, accounting for approximately 40% of cells (Figure 6A–F). These results were further confirmed by an independent Annexin V/PI analysis (data not shown). In addition, observed cell alteration/death in all used models was markedly abrogable upon external addition of zinc, thereby proving that TPEN-dependent chelation of intracellular zinc, but not other elements, was fully responsible for the observed endpoints (Figure 5E and Figure 6E). 

### 2.5. TPEN Induces Caspase-Dependent Mitochondrial Apoptosis

Analyses of mitochondrial cytochrome c, its cytoplasmic translocation, and subsequent caspase-3 activity in TPEN-treated melanoma cells revealed their significant involvement (Figure 7A–C). To this extent, pretreatment of TPEN-exposed cells with the specific caspase-3 inhibitor or with cyclosporine A—the specific inhibitor of cytochrome c—had a dramatic reducing effect on cell damage/death rate (Figure 8A,B).

### 2.6. Loss of Free Intracellular Zinc Has Disparate Effects on Production of Superoxide, Damage of Lysosomal Membrane, and p53 Activity

Superoxide production in TPEN-exposed melanoma cells showed a time-dependent and cell-type-dependent course. Its significantly increased levels were present in exposed Bowes and A375 cells already at 12 h of exposure, with maximum reached at 24 h (Bowes) and 48 h (A375), respectively. On the other hand, in both M1 and M2 cultures, superoxide concentrations remained virtually unchanged during the first 12 h of treatment and gradually reached significant levels at later time intervals only. At all measured time intervals, however, the generation of superoxide in these cultures remained significantly lower than in the employed cell lines (Figure 9A). 

The integrity of the lysosomal membrane in all examined melanoma cells remained unchanged during 12 h of TPEN treatment. Thereafter, lysosomal membrane damage steeply increased in M1 and, in particular, in M2 cultures, whereas in Bowes and A375 cells, while markedly elevated at later treatment intervals, it never reached the extent observed in the examined cell lines (Figure 9B).

Measurement of transcription activity of p53 in TPEN exposed melanoma cell lines and cultures also revealed robust differences. In Bowes cells and M1 cultures, p53 activity increased until 24 h of exposure and then slightly decreased or remained constant. Conversely, in A375 cells and M2 cultures, p53 activity did not change throughout the experiment (Figure 9C). 

### 2.7. Effects of Zinc Reloading on Cell Cycle of Free Zinc Depleted Melanoma Cells

Extant-free zinc-depleted melanoma cells were exposed to a standard growth medium supplemented with 100 μM zinc sulfate to replete their lost zinc stores. Their morphology and behavior were next followed for another 48 h. Generally, free zinc-depleted cells were less extended, their original shape altered to a varying extent, and cytoplasm coarsened (data not shown). Most of them (more than 99%) were not actively cycling, as indicated by Ki-67 negativity. Exposure to zinc supplemented growth medium for 24 h restored original cell morphology to a significant proportion in melanoma Bowes and A375 cells but not in melanoma explant cultures M1 and M2 (not shown). In addition, at 24 h, approximately 15% (Bowes) and 26% (A375) of cells showed Ki-67 positivity, unlike in M1 and M2 cultures, where the positivity remained very low (1–3%). These effects became even more pronounced at 48 h (Figure 10A,B).

### 2.8. Zinc Depletion Induces Senescence in Melanoma Cells with Varying Influence of Zinc Reloading

Due to cell cycle arrest (along with changed morphologies) in extant free zinc-depleted melanoma cells/cultures, the presence of characteristic hallmarks of premature senescence was finally investigated. Expression of beta-galactosidase was initially found in all examined melanoma models, with the positivity ranging from about 8% in the Bowes cell line to up to 16% in the M2 melanoma culture. Proportions of these positive melanoma Bowes and A375 cells did not significantly change during the next 48 h despite their exposure to a standard growth medium supplemented with zinc. On the other hand, beta-galactosidase positivity grew in M2 melanoma cultures, where it reached 40% at 48 h (Figure 11A). Moreover, microscopic examination revealed specific morphotypes in all examined melanoma cells characteristic of premature senescence, too (Figure 11B). Finally, immunoblotting analysis of p21 and p16 markers indicated that, unlike in control melanoma cells grown upon standard cultivation conditions, the expression of both of them was significantly elevated. This increased expression occurred even when the originally free zinc-depleted cells were treated for 48 h in a zinc-supplemented medium. Increased protein p21 and p16 abundancies were more significantly present in M1 and M2 explant cultures (Figure 11C).

## 3. Discussion

Unlike several solid malignancies such as breast cancer, prostate cancer, or pancreatic cancer, where changes in zinc management and overall zinc content, including labile/free zinc pools, are recognized and well researched, the role of zinc in melanoma is less clear, and it is evidenced only partially and indirectly [14,15,16]. Mechanistic in vitro studies, in particular, are scarce to demonstrate how melanoma cells would respond to acute changes in their intracellular free zinc pools. 

In the current study, we used both stabilized melanoma cell lines Bowes and A375 and two melanoma explant cultures established from clinical samples with known low (M1) and high (M2) free zinc. To study the biological effects of acute free zinc chelation, we used the agent TPEN at the concentration of 25 µM, which proved in initial analyses to effectively reduce free zinc levels. TPEN-induced free zinc sequestration further inhibited cell proliferation in both melanoma cell lines as well as in explant cultures and resulted in G1/S cell cycle arrest with subsequent cell damage/death, although its rate differed in a melanoma model-dependent manner. TPEN-treated cells lost their adherence, collapsed, and developed membrane blebs indicative of classical apoptosis, although some of them displayed other features (Figure 4 and Figure 5A). These observations were further confirmed by the presence of active caspase-3 in treated cells and mechanistically verified via specific caspase-3 and cytochrome c pharmacological inhibitors. Such outcomes concur with other reports demonstrating a similar cell cycle arresting and proapoptotic effect of TPEN-related zinc chelation in pancreatic, ovarian, leukemic, osteosarcoma, or neuroblastoma cells [17,18,19,20]. Chelation of intracellular zinc via TPEN or other zinc-binding agents may generally damage and inhibit different types of cancer cells, but the ultimate mechanism behind this activity seems to differ. In this context, various mechanisms have been suggested, including oxidative stress, which independently on TPEN-related zinc-binding activity triggers the signaling axis of NFκB-p53 and leads to mitochondrial apoptosis in lymphoblastic leukemia cells [21]. Similar findings are reported in the case of pancreatic cancer cells where TPEN acted via mitochondrial production of superoxide. In addition, in thus exposed cells, autophagy was inhibited, which authors attributed to TPEN-dependent disruption of the lysosomal membrane [22]. On the other hand, in neuroblastoma cells, TPEN-dependent cell death occurred exclusively via lysosomal disruption without any involved oxidative stress [23]. Pleiotropic effects of TPEN are finally demonstrated in several pancreatic adenocarcinoma lines where this compound induced cell cycle arrest in the G1 phase via increased expression of p57 and p19 inhibitors and elevated ratio of apoptotic/antiapoptotic genes [24]. Our current findings show G1/S phase cell cycle arrest as demonstrated by reduced proportions of S-phase cells in TPEN-exposed cultures. The underlying processes include both increased generation of superoxide as well as lysosomal membrane damage, albeit both processes seem to have a mutually exclusive status in particular cells. Thus, in treated Bowes and A375 cells, a time-dependent buildup of superoxide levels occurred in the absence of a significant lysosomal damage whereas in melanoma samples an opposite situation was noted. Oxidative stress is known to induce lysosomal membrane damage contributing in many cases to the ultimate cell damage and cell death [25], while lysosomal permeabilization may occur in the absence of oxidative damage [23]. Our results indicate that, at least in our employed models, oxidative stress and lysosomal damage related to TPEN appear to be independent and not directly related phenomena, likely reflecting a degree of differences between both groups of models, including individual levels of free zinc.

Zinc is required by numerous cellular proteins, including transcription factors, for their optimal biological functioning. In the case of p53, zinc works to stabilize both its global structure as well its local DNA-recognizing elements [26]. It is nowadays known that loss of zinc such as due to its pharmacologically (i.e., TPEN)-stimulated chelation may suppress p53 nuclear localization and subsequent p53 transcriptional activation [27]. Conversely, the addition of zinc or metallochaperones to mutant p53 could restore wild-type p53 structure and function [28,29]. 

To this extent, TPEN-dependent sequestration of intracellular zinc in A375 melanoma cells bearing wild-type p53 gene status led to almost complete prevention of p53-DNA binding, unlike in Bowes cells (mutant p53) where DNA-binding activity of p53 increased. In the employed melanoma explant cultures, similarly varying results were obtained with M1 and M2 cells whose p53 status, however, unchecked by us, could have reflected their different responses.

While damaging to a considerable proportion of exposed melanoma cells, some of them did not show any sign of morphological changes. Still, their active cycling (both cell lines as well as explant cultures) was arrested, as evidenced by their lack of expression of Ki-67 marker. To gain further insight into the nature of this arrest, we exposed these cells again to standard culture media containing external zinc. As shown in Figure 10, already after 24 h of such exposure, some previously arrested cells started to return to active cycling, which became further evident at 48 h; however, significantly differing rates were recorded in cell lines and melanoma cultures. In parallel with the gradually increasing return of cells into the cell cycle, a growing number of them exhibited premature senescence phenotype. In stabilized cell lines Bowes and A375, most extant cells resumed active cycling while up to 20% of cells showed positivity for premature senescence. Conversely, in melanoma explant cultures, the number of senescent cells steeply grew and reached more than 40%, whereas only a very minor fraction of other cells actively cycled. These differences are likely to be due to a number of variables, including initial intracellular free zinc levels, overall zinc management as well as the differences in individual models (stabilized lines versus primary cultivations). Senescence is known to be induced by a number of factors, with a prominent role in oxidative-stress-mediated damage of important macromolecules (DNA, lipids, proteins) [30]. Disturbed homeostasis of intracellular zinc (zinc loss) linked to reactive oxygen species was reported to associate with premature aging and immunosenescence of normal [31] as well as tumor cells [32]. Our results align with these observations and extend them to melanoma which corresponds to an important role trace elements such as zinc, iron, and copper play in melanoma biology [33]. 

In summary, free zinc loss induced by TPEN in melanoma cells leads to their inhibited proliferation, G1/S cell cycle arrest, damage, and mitochondrial, caspase-dependent apoptosis. This process occurs via stimulated oxidative stress or lysosomal damage in the presence or absence of p53 activity. Moreover, the arrested cycling of melanoma cells could be abrogated to a varying extent by exposure of cells to standard cultivation conditions (cultivation media containing zinc). Concurrently, premature senescence phenotype develops in thus cultivated cells, again in a strongly cell-dependent manner. These results, for the first time, show that acute intracellular zinc depletion leads to rapid apoptosis in some melanoma cells while in others, cell cycle arrest occurs, which may in individual cells develop in premature senescence phenotype with interesting clinical management potential. In the future, these conclusions should be verified on larger sample sets to verify and confirm the robustness of these data as this constitutes one of the limitations of the current study.

## 4. Materials and Methods

### 4.1. Cell Lines

Melanoma cell lines A376 (ATCC, No. CRL—9607, Manassas, VA, USA) and Bowes were maintained in DMEM with 10% fetal bovine serum (Gibco, Prague, Czech Republic), 100 U/mL penicillin, and 100 μg/mL streptomycin. Cells were passaged twice a week (0.05% trypsin/EDTA) and kept in an incubator at 37 °C and 5% CO_2_ atmosphere. Cultures were periodically checked for mycoplasma infection. All experiments were carried out with Bowes and A375 cells in the low passages (5–10).

### 4.2. Explant Melanoma Cultures

Explant melanoma cultures (M1 and M2) were obtained, characterized, and cultivated as described before [34].

### 4.3. Chemicals

Newport Green™ DCF diacetate, MitoSOX™ Red were acquired from Molecular Probes, Inc. (Eugene, OR, USA). TPEN (N,N,N′,N′-tetrakis(2-pyridinylmethyl)-1,2-ethanediamine), acridine orange, cyclosporin A, 3-[(3-cholamidopropyl)dimethylammonio]-1-propanesulfonic acid (CHAPS), EdU (5-ethynyl-2′-deoxyuridine), horseradish peroxidase, Triton-X, dithiotreitol (DTT), propidium iodide and 4’,6-Diamidino-2-Phenylindole (DAPI) were obtained from Sigma-Aldrich (St. Louis, MO, USA). WST-1 was purchased from Roche Diagnostics (Manheim, Germany). Caspase-3 inhibitor z-devd-fmk was from ICN Biomedicals Inc. (Irvine, CA, USA). Primary and secondary antibodies were from Cell Signaling Technology (Danvers, MA, USA). All other chemicals were of the highest analytical grade.

### 4.4. Treatment Conditions

TPEN was dissolved in a serum-free medium and stored until use as a stock solution of 1 mM in a refrigerator (4 °C). Inhibitors and other modulators were used in the following way: z-devd-fmk (caspase-3 inhibitor, 5 µM—added to cells simultaneously with TPEN), cyclosporin A (5 µM—supplemented to cells 30 min before exposure to TPEN).

### 4.5. Proliferation and Viability Assay

Human melanoma cell lines A375 and Bowes and explant human melanoma cultures were seeded in 96 well microtiter plates (Nunclon, Roskilde, Denmark) at the initial seeding density of 8000 cells/well with the first column of wells without cells (blank). At each time interval of measurement, 100 μL of WST-1 was added to each well. The cells were further incubated for 2 h in an incubator. Thereafter, the absorbance was measured using the multiplate reader TECAN SpectraFluor Plus (TECAN Austria GmbH, Grödig, Austria) at 450 (excitation) and 690 nm (emission).

### 4.6. S-Phase Cell Fraction Assay

Proportions of S-phase cells were determined by EdU-Click 488 assay (Sigma-Aldrich, St. Louis, MO, USA). Briefly, human melanoma cell lines A375 and Bowes and explant human melanoma cultures were seeded in 96 well microtiter plates (Nunclon, Roskilde, Denmark) at the initial seeding density of 6000 cells/well. At regular time intervals, cells were washed with PBS, and the medium with EdU (10 μM) was added to each well for 2 h. Thereafter, cells were washed with PBS, fixed with paraformaldehyde (15 min), and permeabilized (0.5% Triton-X, 20 min, 25 °C). After repeated washing with PBS and postlabeling with DAPI, EdU buffer additive was added to cells (30 min, dark, 25 °C), and their specific S-phase fluorescence along visualized, recorded, and analyzed by Cell scoring module of MetaXpress^®^ Image Acquisition and Analysis Software. 

### 4.7. Intracellular Zinc Concentrations

The trypsined and rinsed melanoma cells were harvested (approximate 0.2 g of wet mass), placed into quartz vessels, and digested with 3.5 mL 60% nitric acid. Thereafter, sealed vessels were microwave oven heated, rinsed in ultrapure water, and aliquots diluted 1:10 prior to zinc concentration analysis with the inductively coupled plasma emission spectrometer MSD 5972 (Agilent Technologies, Waldbronn, Germany) as based on [35,36]. Prior to analysis, aliquots of the cell samples were assayed for protein content using a BCA assay (Bicinchoninic acid kit for protein determination, Sigma-Aldrich, Prague, Czech Republic). Changes in total intracellular zinc content were expressed as µg of zinc/mg of protein.

Free intracellular zinc levels in the assayed melanoma cells were determined microfluorimetrically based on the published protocol [37]. The cells grown in black-bottom 96-well plates were incubated with Newport Green diacetate (5 μM in PBS, dark, 30 min at 37 °C). Fluorescence intensity was determined by the multiplate reader TECAN SpectraFluor Plus (TECAN Austria GmbH, Grödig, Austria). The results in relative light units were obtained from the raw data minus reagent blank, with changes expressed as a percentage of controls.

### 4.8. Superoxide Production

Untreated and TPEN-exposed melanoma cells were seeded and grown in 96-well plates with black bottom. At individual time intervals, cells were incubated with MitoSOX™ Red solution (5 µM, 20 min, 37 °C), rinsed in warm medium, and the specific fluorescence reflecting superoxide was analyzed by Cell scoring module of MetaXpress^®^ Image Acquisition and Analysis Software. Results were expressed as percentage of cells positive for superoxide ion.

### 4.9. Lysosomal Membrane Damage

Untreated and TPEN-exposed melanoma cells grown in 96-well plates were at given time intervals rinsed, incubated with acridine orange (5 μM, 15 min, 37 °C) and its cytoplasmic redistribution was measured fluorimetrically (TECAN SpectraFluorPlus, TECAN Austria GmbH, Grödig, Austria). Lysosomal membrane damage was expressed as an increase in diffuse cytosolic green fluorescence by acridine orange released from lysosomes in arbitrary units.

### 4.10. P53 DNA-Binding Assay

P53 DNA-binding activity in control and TPEN-exposed melanoma cells was measured using p53 transcription factor assay kit (Cayman Europe, Tallinn, Estonia) as per instructions of manufacturer. The DNA binding reflecting transcription activity of p53 in cultures was determined spectrophotometrically at 450 nm and expressed as percentage of control. 

### 4.11. Dynamic Morphology—Cell Damage/Death/Senescence

Untreated and TPEN-treated melanoma cells were seeded into plastic tissue-culture dishes with glass bottom. Their morphology and behavior were observed in a time-lapse imaging system BioStation IM (Nikon, Prague, Czech Republic). Recording was carried out in both multipoint and multichannel time-lapse modes and upon a range of magnifications. Obtained sequences were software (NIS Elements AR 3.20 (Nikon, Prague, Czech Republic)) processed and analyzed with selection of typical frames depicting morphology of individual cells at particular time intervals. The rate of cell damage/death was expressed as percentage of cells with changed morphology as compared to control cell appearance. The rate of senescence was expressed as percentage of cells with senescent morphology as compared to control cell appearance.

### 4.12. Apoptosis

Stabilized melanoma cell lines A375 and Bowes, as well as melanoma explant cultures (untreated and TPEN-treated), were grown on coverslips. At given time intervals, cells were rinsed in PBS, fixed in 4% paraformaldehyde, and permeabilized with cold methanol/Triton-X in 5% BSA. Fixed and rinsed cells were incubated with anti-cleaved caspase-3 (1:100) at 4 °C for 1 h. Following several rounds of rinsing with cold PBS (each 5 min, 25 °C), FITC-conjugated goat anti-rabbit secondary antibody was added for 1 h at 4 °C. Thereafter, samples were rinsed in cold distilled water, counterstained with DAPI, and mounted. Fluorescent images were obtained with Nikon Eclipse Ni microscope (Nikon, Prague, Czech Republic), with numbers of cells positive for cleaved caspase-3 determined manually using the multidimensional analysis module of NIS Elements AR software. The results were correlated to morphological apoptotic phenotype and expressed as % of apoptotic cells.

### 4.13. Cytochrome Release Assay

Untreated and TPEN-treated melanoma cells grown in cultivation flasks were harvested at specific time intervals, and mitochondrial and cytosolic extracts were prepared using the Mitochondria Fractionation Kit (Active Motif, Rixensart, Belgium). Quantitation of cytochrome c in prepared lysates was determined by FunctionElisa™ cytochrome c kit (Active Motif, Rixensart, Belgium) using spectrophotometrical measurement at 450 nm with a scanning multiwell spectrophotometer Titertek Multiscan MCC/340 (ICN Biochemicals, Frankfurt, Germany). Data were expressed as increase in absorbance at 450 nm/μg of lysate/well. 

### 4.14. Caspase-3 Activity

Untreated and TPEN-treated melanoma cells grown in tissue culture flasks were harvested by centrifugation (600× *g*, 5 min, JOUAN MR 22, Trigon, Prague, Czech Republic) at specific time points. Next, they were lysed on ice for 20 min in a lysis buffer (50 mM HEPES, 5 mM CHAPS and 5 mM DTT). The lysates were centrifuged at 14,000× *g*, 10 min, 4 °C, the supernatants were collected and stored at −80 °C. The enzyme activity was measured in a 96-well microplate using a kinetic fluorometric method based on the hydrolysis of the fluorogenic caspase-specific substrate (DEVD-AFC at 37 °C, 1 h). Specific inhibitor of caspase-3 (z-DEVD-fmk) at concentration of 5 µM was used to confirm the specificity of the cleavage reaction. Fluorescence was recorded at 460/40 nm after excitation at 360/40 nm using TECAN SpectraFluor Plus (TECAN Austria GmbH, Grödig, Austria). The results are shown as fold increase in activity relative to untreated cells. 

### 4.15. Cell Cycling

Stabilized melanoma cell lines A375 and Bowes, as well as melanoma explant cultures (untreated and TPEN-treated), were grown on coverslips. At individual time intervals, treated and control cells were processed, and their Ki-67 positivity was determined via immunofluorescence as detailed in the previous section using primary antibody anti-Ki-67 (1:100) and secondary TRITC-conjugated goat anti-mouse. The results were expressed as percentage of cells positive for Ki-67.

### 4.16. Cellular Senescence Assay

Treated and control colonic fibroblast cultures were at chosen time intervals washed with PBS and fixed with 2% formaldehyde, rinsed three times with phosphate saline buffer (PBS) and incubated overnight (37 °C) with freshly prepared SA-β-gal staining solution containing 1 mg/mL 5-bromo-4-chloro-3-indolyl β-D-galactopyranoside (X-gal) (Calbiochem, EMD Biosciences, Inc., La Jolla, CA, USA), 5 mM potassium ferrocyanide, 5 mM potassium ferricyanide, 150 mM NaCl, 2 mM MgCl_2_ and 40 mM citric acid titrated to pH 6.0. Next, specimens were washed with distilled H_2_O, dehydrated, mounted, and examined under a bright field microscope Nikon Eclipse E 400 (Nikon Corporation, Kanagawa, Japan) with the digital color matrix camera COOL 1300 (VDS, Vosskűhler, Germany). SA-β-gal positivity was semi-automatically analyzed by the software NIS Elements AR 3.20 (Nikon, Prague, Czech Republic) in at least 1000 cells per sample. The results were expressed as percentage of senescent cells. 

### 4.17. Immunoblotting

Treated and control human melanoma cells in cultivation flasks were washed with PBS and harvested at different time intervals in ice-cold lysis buffer (150 mM NaCl, 10% glycerol, 1% n-octyl-β-D-glucopyranoside, 1% Triton X-100, 50 mM NaF, 50 mM Tris/HCl, 2 mM EDTA, 2 mM EGTA, 50 mM NaF, 1 mM sodium orthovanadate, Complete TMMini). The cell lysates were boiled for 5 min/95 °C in SDS sample buffer (Tris-HCl pH 6.81, 2-mercaptoethanol, 10% glycerol, SDS, 0.1% bromphenol blue) and loaded onto a 10% SDS/polyacrylamide gel. Each lysate contained equal amount of protein (15 μg) as determined by BCA assay. After electrophoresis, proteins were transferred to a PVDF (Polyvinylidene fluoride) membrane (200 V, 75 min) and blocked (1 h, at 25 °C) with 5% nonfat dry milk in TBST (Tris-Buffered Saline plus 0.05%Tween-20). Membranes were incubated with primary antibodies (monoclonal mouse anti-p21, 1:1000, monoclonal mouse anti-p16, 1:500, monoclonal mouse anti-β-actin, 1:100) at 4 °C overnight, followed by five 6 min washes in TBST. Next, the blots were incubated with secondary peroxidase-conjugated antibodies (1:1000, 1 h, 25 °C), washed with TBST, and the signal was developed with a chemiluminescence (ECL) detection kit (Boehringer Mannheim-Roche, Basel, Switzerland). β-actin was used as the loading control. Density of protein-specific signals was evaluated by GelQuant 2.7 software (DNR Bio-Imaging Systems, Jerusalem, Israel).

### 4.18. Statistics

Statistical analysis was carried out with the statistical program GraphPad Prism (GraphPad Software version 7.0, Inc., San Diego, CA, USA). We used a one-way-Anova test with Dunnett’s post-test for multiple comparisons. Results were compared with control samples, with means considered significant at *p* < 0.05.

## Figures and Tables

**Figure 1 ijms-23-08312-f001:**
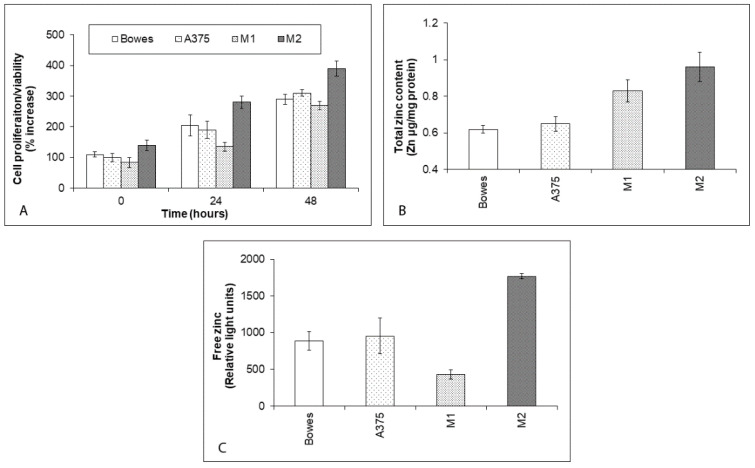
The proliferation and zinc content in human melanoma Bowes and A375 cell lines and explant melanoma cultures. Human melanoma cell lines Bowes and A375 and established explant human melanoma cultures (M1 and M2) were maintained upon standard laboratory conditions and their proliferation was determined by (**A**) colorimetric WST-1 assay. Values represent means ± SD of at least three experiments. (**B**) Total zinc content in human melanoma cell lines Bowes and A375 and established explant human melanoma cultures (M1 and M2) as determined by absorption spectrometry. (**C**) Free (labile) zinc in human melanoma cell lines Bowes and A375 and established explant human melanoma cultures (labeled M1 and M2) as measured by microfluorimetry. Values represent means ± SD of at least three experiments.

**Figure 2 ijms-23-08312-f002:**
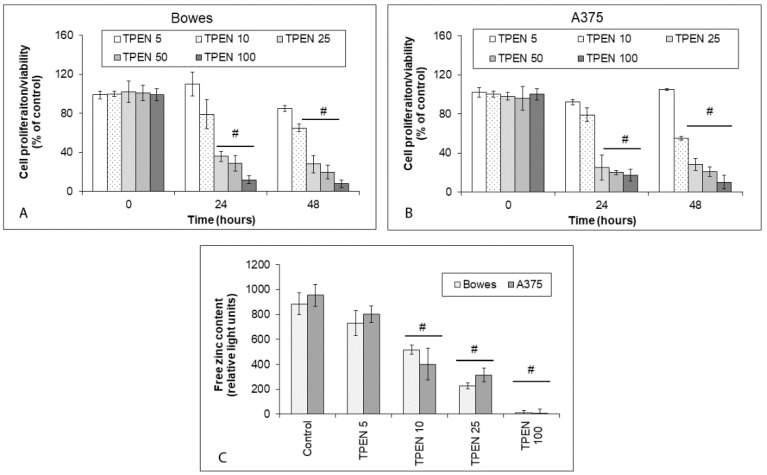
The proliferation and free zinc in human melanoma Bowes and A375 cell lines exposed to intracellular zinc chelating TPEN. Cells were exposed to TPEN at the concentration range of 5–100 µM during 48 h and their proliferation was determined by (**A**,**B**) colorimetric WST-1 assay. Values represent means ± SD of at least three experiments. *p* < 0.05 # Significantly lower than control at the same treatment interval with a one-way-Anova test and Dunnett’s post test for multiple comparisons. (**C**) Free (labile) zinc in melanoma cells as measured by microfluorimetry. Values represent means ± SD of at least three experiments. *p* < 0.05 # Significantly lower than control cells at the same treatment interval with a one-way-Anova test and Dunnett’s post test for multiple comparisons.

**Figure 3 ijms-23-08312-f003:**
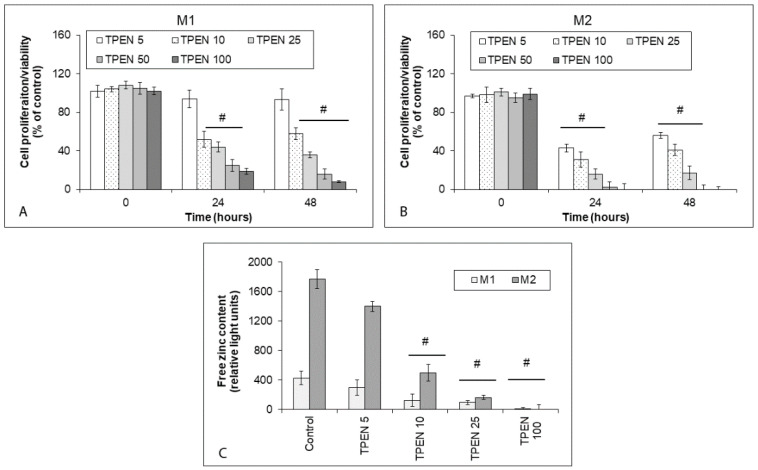
The proliferation and free zinc in human melanoma explant cultures M1 and M2 exposed to intracellular zinc chelating TPEN. Explant cultures were exposed to TPEN at the concentration range of 5–100 µM during 48 h and their proliferation was determined by (**A**,**B**) colorimetric WST-1 assay. Values represent means ± SD of at least three experiments. *p* < 0.05 # Significantly lower than control cells at the same treatment interval with a one-way-Anova test and Dunnett’s post test for multiple comparisons. (**C**) Free zinc in melanoma cultures as measured by microfluorimetry. Values represent means ± SD of at least three experiments. *p* < 0.05 # Significantly lower than control cells at the same treatment interval with a one-way-Anova test and Dunnett’s post test for multiple comparisons.

**Figure 4 ijms-23-08312-f004:**
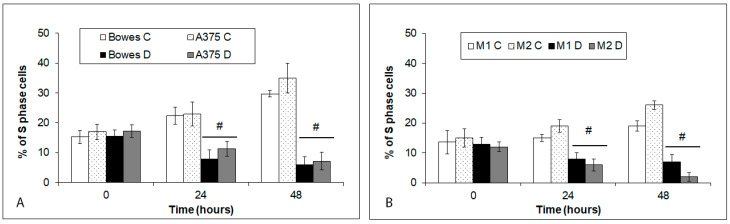
The S-phase fraction of human melanoma cells. (**A**) Bowes and A375 cell lines and (**B**) explant cultures M1 (low free zinc) and M2 (high free zinc) were exposed to 25 µM TPEN and proportions of their cells in S phase were determined with EdU assay. Values represent means ± SD of at least three experiments. *p* < 0.05 # Significantly lower than control cells at the same treatment interval with a one-way-Anova test and Dunnett’s post test for multiple comparisons.

**Figure 5 ijms-23-08312-f005:**
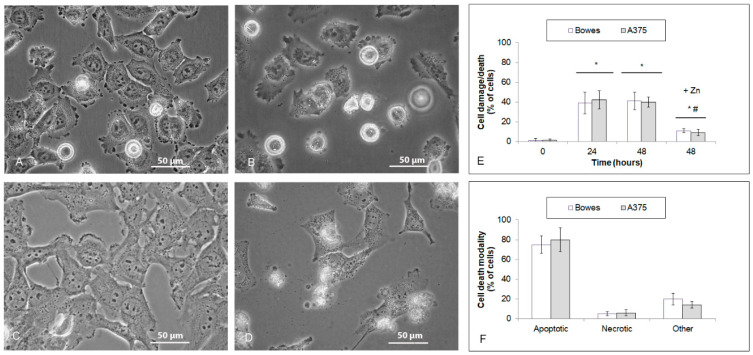
Cell damage/death of human melanoma cells Bowes and A375 exposed to 25 μM TPEN during 48 h. Cells were treated with TPEN alone or with zinc sulfate (100 μM) and their effect on cell damage/death was evaluated by morphometric analysis of cell micrographs obtained from time-lapse microscopy (**A**–**E**) and propidium iodide (**F**). Results represent means ± SD of four experiments. (**A**–**D**) Phase contrast microscopy, 600×. (**E**) *p* < 0.05 * Significantly higher compared to the beginning of treatment, *p* < 0.05 # Significantly lower than TPEN only treated cells at 48 h with one-way-Anova test and Dunnett’s post test for multiple comparisons.

**Figure 6 ijms-23-08312-f006:**
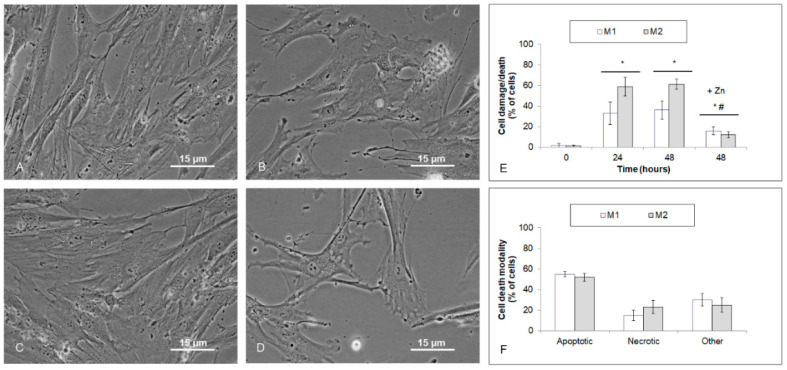
Cell damage/death of human melanoma explant cultures M1 (low free zinc) and M2 (high free zinc) exposed to 25 μM TPEN during 48 h. Cells were treated with TPEN alone or with zinc sulfate (100 μM) and their effect on cell damage/death was evaluated by morphometric analysis of micrographs obtained from time-lapse microscopy (**A**–**E**) and propidium iodide (**F**). Results represent means ± SD of four experiments. (**A**–**D**) Phase contrast microscopy, 600×. (**E**) *p* < 0.05 * Significantly higher compared to the beginning of treatment, *p* < 0.05 # Significantly lower than TPEN only treated cells at 48 h with one-way-Anova test and Dunnett’s post test for multiple comparisons.

**Figure 7 ijms-23-08312-f007:**
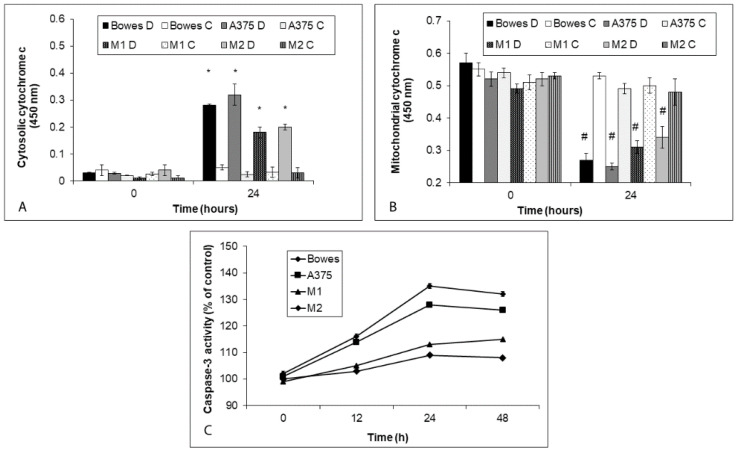
Cytoplasmic translocation of mitochondrial cytochrome c and activation of caspase-3 in human melanoma cells exposed to 25 μM TPEN during 48 h. C = control cells. D = zinc depleted cells. Changes in (**A**) cytosolic and (**B**) mitochondrial cytochrome c content and (**C**) caspase-3 activity in treated cells were determined as described in Materials and methods section. Results represent means ± SD of five experiments. *p* < 0.05 * Significantly higher compared to the beginning of treatment of the same cells, *p* < 0.05 # Significantly lower compared to untreated cells at the same time interval with one-way-Anova test and Dunnett’s post test for multiple comparisons.

**Figure 8 ijms-23-08312-f008:**
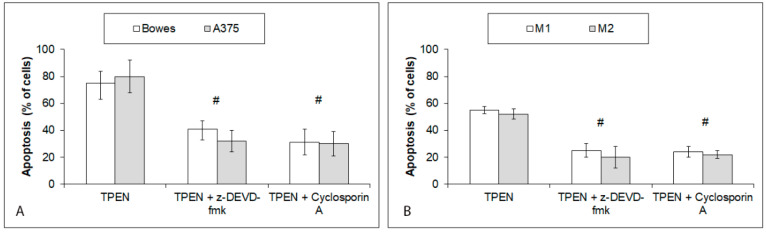
Effects of specific caspase-3 or cytochrome c mitochondrial release inhibition on 25 μM TPEN induced apoptosis of human melanoma cells during 48 h. Cells of (**A**) stabilized cell lines Bowes and A375 and (**B**) melanoma explant cultures M1 (low free zinc) and M2 (high free zinc). Treatment of cells with TPEN and inhibitors and quantification of apoptosis were carried out as described in Materials and methods section. Results represent means ± SD of five experiments. *p* < 0.05 # Significantly lower than TPEN only treated cells at 48 h time interval with one-way-Anova test and Dunnett’s post test for multiple comparisons.

**Figure 9 ijms-23-08312-f009:**
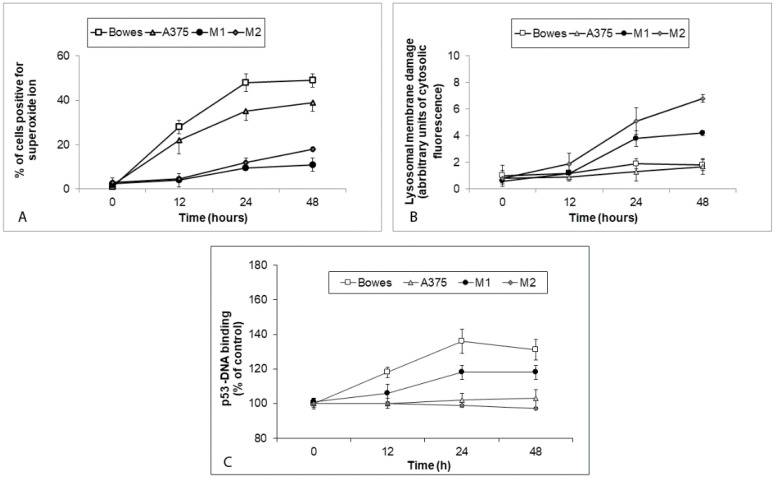
Generation of superoxide, lysosomal membrane damage and p53 activity in human melanoma cells Bowes and A375 and melanoma explant cultures M1 (low free zinc) and M2 (high free zinc) exposed to 25 μM TPEN during 48 h. In exposed cells (**A**) superoxide levels (**B**) lysosomal membrane damage and (**C**) changes in p53 activity were measured as described in Materials and methods section. Results represent means ± SD of at least three independent experiments.

**Figure 10 ijms-23-08312-f010:**
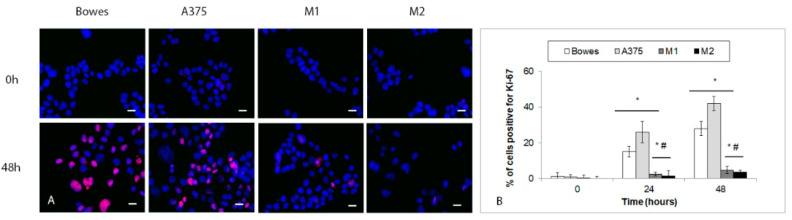
Cell cycle arrest in free zinc depleted human melanoma cells Bowes and A375 and melanoma explant cultures M1 (low free zinc) and M2 (high free zinc) treated to standard growth medium supplemented with 100 μM zinc sulfate during 48 h. In exposed cells Ki-67 positivity was determined by immunofluorescence. (**A**) Fluorescent microscopy, 200×. Scale bar 2 μm. Blue—DAPI; red—Ki-67. Results represent means ± SD of at least three independent experiments. (**B**) Quantification of Ki-67 positivity. Results represent means ± SD of at least three independent experiments. *p* < 0.05 * Significantly higher compared to the beginning of treatment, *p* < 0.05 # Significantly lower than Bowes and A375 cells at the same time interval with one-way-Anova test and Dunnett’s post test for multiple comparisons.

**Figure 11 ijms-23-08312-f011:**
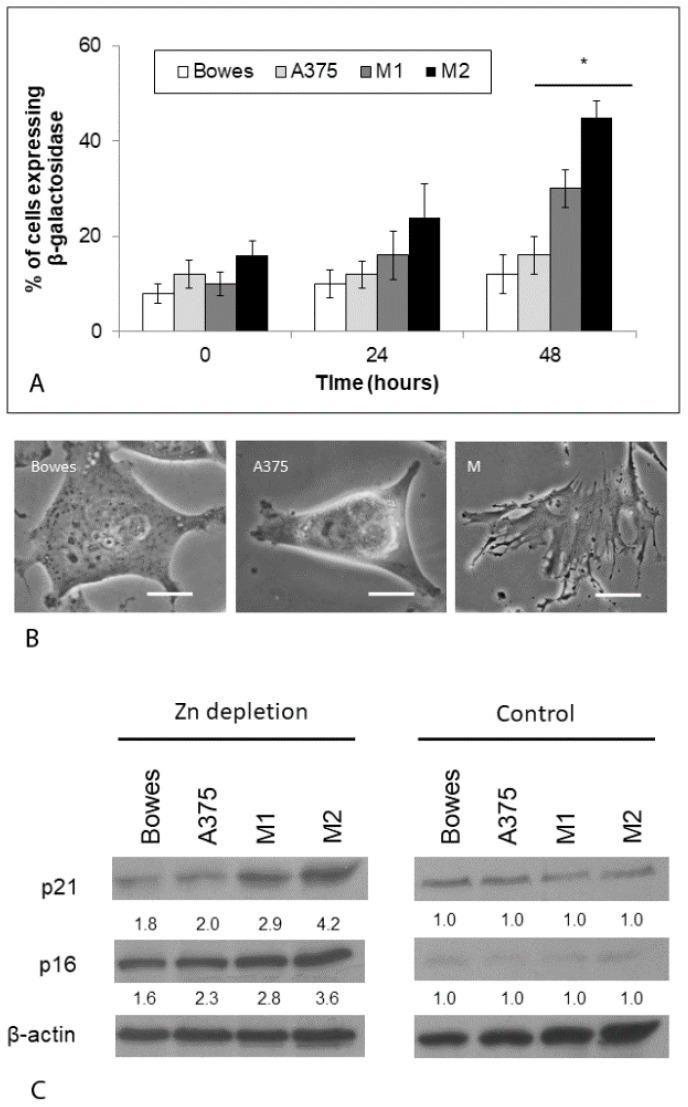
Premature senescence in free zinc depleted human melanoma cells Bowes and A375 and melanoma explant cultures M1 (low free zinc) and M2 (high free zinc) before and after exposure to standard cultivation medium supplemented with 100 μM zinc sulfate during 48 h. (**A**) Beta galactosidase positivity was determined microscopically. Results represent means ± SD of at least three independent experiments. *p* < 0.05 * Significantly higher compared to the beginning of treatment with one-way-Anova test and Dunnett’s post test for multiple comparisons. (**B**) Phase contrast microscopy, 600×, scale bar 2 μm. (**C**) Expression of p21 and p16 proteins was determined by immunoblotting. Control—cells cultivated under standard, zinc adequate conditions (48 h). Zn-depletion—cells with original free zinc depleted and then cultivated for 48 h with standard growth medium supplemented with 100 μM zinc sulfate. The loading was normalized to β-actin and quantitative analysis of bands was carried out by GelQuant Ver 2.7 software (DNR Bio-Imaging Systems, Jerusalem, Israel). The numbers in the blot image refer to fold increase or decrease in the density of particular protein compared to the density of the same protein in the same cells at the beginning of treatment. Shown is one typical result of at least four experiments.

## Data Availability

Data is contained within the article.

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
