# Peer review of "Induced Zinc Loss Produces Heterogenous Biological Responses in Melanoma Cells"

_ijms, 2022, doi:10.3390/ijms23158312_

Round 1

Reviewer 1 Report

Reviewer Comments

The research article by Emil Rudolf and Kamil Rudolf titled “Induced zinc loss produces heterogenous biological responses in melanoma cells” is an interesting study. However, some of my major concerns are

1.     In the result section 3.1, the authors have mentioned “Analyses of mitochondrial cytochrome c cytoplasmic translocation and subsequent caspase-3 activity in TPEN-treated melanoma cells revealed their significant involvement (data not shown). I request the authors to provide the data.

2.     I request the authors to perform flow cytometry for after treatment with TPEN at different dose range for all the cell lines mentioned in the data for accurately proving apoptosis.

3.     At what stage of the cell cycle did the cells undergo cell cycle arrest. I also request the authors to perform flow cytometric analysis for evaluation of cell cycle arrest as well as apoptosis.

Minor comments

1.     Make changes to section 2.1, A376 melanoma cell line is it A375 cell line?

2.     Please mention the passage number of the cell line A375 in section 2.1.

Reviewer 2 Report

Manuscript is well and clearly written. The paper is carefully edited. Introduction gives relevant background for the investigation.  Aim of the study is clearly highlighted.

I have some suggestions/questions  for Authors:

Major comment:

Without detailed description of procedure for free intracellular zinc, it is difficult to assess what form of Zn was really measure. Why Author think that in such test only “free” Zn reacts with reagent? Moreover, the term “free zinc content” is incorrect because Authors showed only relative light units.

Minor comments

1) Add some experimental detail for ICP analysis (spectral line, apparatus parameters…) or add appropriate reference. Why two different methods were used for Zn analysis?

2) The amount of sample (0.35 Ml) is too low for ICP analysis. Was the sample diluted?

3) “cells were dissolved in 0.35 ml 0.8% nitric acid” – the conditions of mineralization are very mild. Was the sample heated? On what basis the conditions were chosen (add reference)?

4) Viability of cells (Fig. 1a,) should be expressed as percentage value comparing to control (the same comment for Figs. 2a,3)

5) Fig. 1: Fig.1a should be separate figure because it is hardly visible (the same comment for Fig. 2a, 3)

6) Figure legend should be reedited, e.g. some information such as the description mechanism of action WST-1 is unnecessary

7)The other Figures: the quality of figures should be improved.

8) Abbreviation “ TPEN” should be explained

Round 2

Reviewer 1 Report

Revision review

1.     Although the authors did not perform the flow cytometry analysis for apoptosis and cell cycle arrest. However, the authors have performed an experiment for apoptosis which is also okay. I have no objection to the manuscript and I leave it to the reviewers for acceptance of the manuscript. 

Author Response

Authors thank the reviewer for the positive acknolwedgement of responses to hs/her earlier comments and questions and the final decision.

Reviewer 2 Report

Authors addressed all my comments and provided necessary explanation. The appropriate corrections have been made.

However manuscript should be prepared using IJMS template.

Author Response

(The authors gave the same response as above.)
